# Rational Optimization of Cathode Composites for Sulfide-Based All-Solid-State Batteries

**DOI:** 10.3390/nano13020327

**Published:** 2023-01-12

**Authors:** Artur Tron, Raad Hamid, Ningxin Zhang, Alexander Beutl

**Affiliations:** AIT Austrian Institute of Technology GmbH, Center for Low-Emission Transport, Battery Technologies, Giefinggasse 2, 1210 Vienna, Austria

**Keywords:** Li_6_PS_5_Cl, argyrodite, all-solid-state battery, processing, lithium battery

## Abstract

All-solid-state lithium-ion batteries with argyrodite solid electrolytes have been developed to attain high conductivities of 10^−3^ S cm^−1^ in studies aiming at fast ionic conductivity of electrolytes. However, no matter how high the ionic conductivity of the electrolyte, the design of the cathode composite is often the bottleneck for high performance. Thus, optimization of the composite cathode formulation is of utmost importance. Unfortunately, many reports limit their studies to only a few parameters of the whole electrode formulation. In addition, different measurement setups and testing conditions employed for all-solid-state batteries make a comparison of results from mutually independent studies quite difficult. Therefore, a detailed investigation on different key parameters for preparation of cathodes employed in all-solid-state batteries is presented here. Employing a rational approach for optimization of composite cathodes using solid sulfide electrolytes elucidated the influence of different parameters on the cycling performance. First, powder electrodes made without binders are investigated to optimize several parameters, including the active materials’ particle morphology, the nature and amount of the conductive additive, the particle size of the solid electrolyte, as well as the active material-to-solid electrolyte ratio. Finally, cast electrodes are examined to determine the influence of a binder on cycling performance.

## 1. Introduction

Sulfide-based solid electrolytes showing high ionic conductivities of more than 10^−3^ S cm^−1^ [1,2,3] have become commercially available in recent years, and new materials are continuously developed [4,5]. Concomitantly, the focus of research has diversified, and the race for higher ionic conductivities has become only one of many topics of interest, including the investigation of the cathode–electrolyte interface/phase [6], processing of electrolyte films [7,8,9], and upscaling [10,11]. The interface between active materials and solid electrolyte particles in composite electrodes as well as their processability have drawn an increasing amount of attention [12,13,14,15] and have been identified as one of the bottlenecks for high performing all-solid-state batteries [16]. Engineering highly stable interfaces with good electronic as well as ionic conductivity is one of the main requirements for highly performing all-solid-state battery cells [12,17]. Nevertheless, the parameters available for optimization are not well understood yet. In most studies, only a few parameters are considered for the optimization of the composite cathodes [13,18,19,20], or they are conducted only theoretically, without any experimental validation [21,22]. Furthermore, many different experimental setups as well as testing conditions are established in the literature [23,24], which makes a direct comparison of data difficult. Even contradictory results are reported, which impedes a rational design of composite electrodes. For example, Ates et al. [18] reported on the use of different conducting additives for composite cathodes and concluded that materials, such as carbon black (C65), increase decomposition reactions at the electrode–electrolyte interface and should be consequently avoided. Thus, the use of alternative materials, such as vapor-grown carbon fibers, has been suggested. In contrast, C65 was used in other reported works [13,20,25,26] and often showed the best performance among the tested additives.

The aim of this work is to provide a rational approach for the optimization of composite cathodes using solid sulfide electrolytes. The influence of different parameters available for composite cathode optimization is regarded and elucidated using a harmonized testing protocol.

First, powder electrodes made without binders were investigated to optimize several parameters, including the active material particle morphology, the nature and amount of conductive additives, the particle sizes of the solid electrolytes, and the active material- to-solid electrolyte ratio. Finally, cast electrodes are examined to further explore the influence of a binder on cycling performance.

## 2. Materials and Methods

Materials—Two kinds of argyrodite materials (Li_6_PS_5_Cl) were used in this work: a commercially available material purchased from NEI Corp. and a material in development kindly provided by Solvay. The two Li_6_PS_5_Cl materials mainly differed in their particle sizes. The material from NEI showed large agglomerates of several tens of µm and will be referred to as lp-Li_6_PS_5_Cl (lp for large particles), whereas the material from Solvay showed smaller particles in the lower µm range and will be referred to as sp-Li_6_PS_5_Cl (sp for small particles). Both materials were used as received unless otherwise stated.

Two different NMC811 materials were used: one commercially available, purchased from NEI Corp., and another material in development, kindly provided by Umicore. The NEI material is LiNbO_3_-coated NMC811 polycrystalline powder (p-NMC811 for polycrystalline), whereas the Umicore one is a LiNbO_3_-coated NMC811 single crystalline powder (s-NMC811 for single crystalline). The two materials were used as received unless otherwise stated.

The selected solvents, p-xylene (anhydrous, ≥99%, Sigma Aldrich), dibromo methane (99%, Sigma Aldrich), and ethyl acetate (anhydrous, 99.8%, Sigma Aldrich) were dried under molecular sieve (0.4 nm, Merck) using a ratio of 1:20 (wt./vol.), i.e., 1 g of molecular sieve to 20 mL of solvent. Poly(acrylonitrile-co-butadiene) (NBR, Perbunan 1846F, Arlanxeo) and poly(butyl methacrylate) (PBMA; Mw ~211000, Sigma Aldrich) were used as a binder for the electrolyte films, whereas poly(styrene-butadiene-styrene) (SBS, Mw ~169000, Sigma Aldrich) was used as a binder for composite cathode films. All binders were dried at 60 °C under vacuum for one day before use.

Processing and manufacturing of electrolyte films—Electrolyte films were prepared using a wet-chemical process. All steps were conducted in an Ar-filled glovebox (H_2_O < 0.1 ppm; O_2_ < 0.1 ppm) unless otherwise stated. First, NBR (9 wt.%) was dissolved in a mixture of dibromo methane (DBM) and ethyl acetate (EtAc) in a 47:53 wt./wt. ratio. The solvents and ratio had been previously investigated, and this formulation yielded the best performance. The solution was mixed using a PTFE-coated stirring bar. Appropriate weights of Li_6_PS_5_Cl and the binder solution were put into a polypropylene (PP) container. Then, the solid content of the slurry was adjusted by addition of the DBM+EtAc solvent mixture to around 50 wt.%. The PP container was closed and sealed with Parafilm^®^ and duct tape. It was shuttled out of the glovebox and mixed using a planetary centrifugal mixer (THINKY ARE 250) for 3 min at 2000 rpm followed by a degassing step for 2 min at 1000 rpm. The electrolyte slurry was again shuttled into the glovebox and applied onto a PTFE foil (50 µm) by tape casting. The gap size of the doctor blade was set to 200 µm. The cast films were further dried at room temperature for 24 h. Finally, the film was lifted off from the substrate and circular shapes of 16 mm diameter and thicknesses of ~100 µm were punched from the film.

Preparation of composite and conventional electrodes—All steps were conducted in an Ar filled glovebox (H_2_O < 0.1 ppm; O_2_ < 0.1 ppm) unless otherwise stated. Composite cathodes were prepared by weighing the components, i.e., active material (AM), solid electrolyte (SE), and conductive additive (CA), and manually mixing them with an agate mortar and pestle until a homogeneous mixture was obtained. The sp-Li_6_PS_5_Cl was used unless otherwise stated. For cast electrodes, pre-dissolved binder solutions were added to the pre-mixed powders, and the solid content was adjusted by adding the solvent. The electrode slurry was mixed in a planetary centrifugal mixer, following the same procedure for the electrolyte slurries. Finally, the electrodes were tape cast onto Al foil. The doctor blade was set to yield electrode loadings of around 1 mAh/cm^2^. Drying of the electrodes was conducted at room temperature for 24 h. The thus obtained electrode sheets were further cut into circular electrodes with 15 mm diameter.

Conventional electrodes using PVdF as a binder were also prepared for measurements in organic liquid electrolytes. These were prepared inside a dry room with a dew point of approximately −50 °C. Electrodes were cast with cathode slurries using NMP as a solvent and an AM:CA:PVdF ratio of 90:05:05 (wt./wt.). The slurry was applied to an Al foil and further dried at 120 °C for 16h under vacuum.

Electrochemical testing of the cathode composites—For electrochemical testing, In foil (Sigma Aldrich, thickness 0.127 mm, 99.99%) was used as a counter electrode instead of Li in order to avoid internal short circuiting of the cell by dendrite formation through the electrolyte layer. In electrodes were prepared by pressing circular pieces (9 mm diameter) of the foil onto steel spacers (16 mm diameter, 500 µm thickness) at 300 MPa. The thickness of the In was thus reduced to ~40–50 µm, and the diameter increased to 14–15 mm. The bare side of the In was covered with a pouch foil (Dai Nippon Printing, D-EL408PH(3)S-250) during pressing to avoid adhesion of the foil to the steel plungers of the hydraulic press. The capacity of the In electrodes was well above 10 mAh (taking only the formation of LiIn into account), i.e., the cathode capacity was always the limiting parameter. The In electrodes were placed into a pressure cell further described in [24]. Three layers of the 100 µm thick electrolyte films were added, and the assembly was pressed at 300 MPa for 5 min. Finally, either the powder or cast composite cathodes were placed on top, aiming for a loading of 1 mAh/cm^2^ (i.e., around 9–10 mg/cm^2^). The whole cell setup was again pressed at 300 MPa for 5 min to yield the final assembly. Then, the cell was placed into the cell holder and a pressure of 10 MPa was applied during the measurements. Galvanostatic cycling with potential limitation measurements was conducted with a Gamry Interface 1010 E. Current densities of 0.05 mA/cm^2^ (C/20) and potential limits of 3.7 V and 2.4 V vs. In/LiIn (4.3 V and 3.0 V vs. Li/Li^+^) were applied. Electrochemical impedance spectroscopy (EIS) was performed from 1 MHz to 1 Hz with an amplitude of 10 mV in PEIS mode.

Conventional electrodes were tested using an organic liquid electrolyte (1M LiPF_6_ in EC:DEC = 1:1 vol./vol.). Electrodes were cut into circular shapes of 15 mm diameter and put into a 2016 coin cell casing. A Celgard 2400 film was used as a separator, and a Li foil (15.6 mm diameter, 250 µm thickness) was used as a counter electrode. The cells were tested in a potential range of 3.0–4.3 V vs. the counter electrode.

The electronic conductivity of electrodes was determined by DC polarization using steps of 10 mV in the range of 10–50 mV. The data were fitted with a linear function, and the slope of it yields the electronic conductivity [27]. All measurements were conducted at pressures between 10 and 300 MPa to determine the pressure dependence of the electronic conductivity.

Powder X-ray diffraction and Scanning electron microscopy—Powder X-ray diffraction (XRD) measurements were performed on a PANalytical X’Pert Pro instrument using Bragg–Brentano geometry. A Cu tube with an Ni filter was used as an X-ray source. Samples for XRD were prepared and sealed in an Ar-filled glovebox using a sample holder with a polymer cap. The polymer cap shows a broad peak around 20°2θ, which can be observed in all diffraction patterns. Measurements were conducted using a 2θ range of 5–80°.

A ZEISS Supra 40 electron microscope was used for scanning electron microscopy (SEM) studies. An acceleration voltage of 3 kV was used for all samples. The samples were mounted on sample holders inside an Ar-filled glovebox and transferred to the SEM within a sealed container. The container was opened only for mounting the samples to the SEM. Thus, contamination due to adsorbed moisture from the surrounding air could be limited to a minimum.

## 3. Results

The approach for the optimization as well as a scheme of the cell assemblies and testing conditions are summarized in Figure 1. First, LiNbO_3_-coated single-crystalline s-NMC811 is compared with an equivalent polycrystalline material, p-NMC811. Furthermore, the influence of the particle size of the Li_6_PS_5_Cl electrolyte component on the cathode was determined. Subsequently, two different conducting additives, i.e., C65 and vapor grown carbon fibers (VGCF), for which contradictory findings were found in the literature [18,20], were tested. Moreover, the active material-to-solid electrolyte (AM:SE) ratio as well as the mixing procedure of the components (conventional vs. sequential) was optimized. Finally, the influence of the binder on the electrochemical performance of the cell was determined. Additionally, some aspects for electrode preparation, especially regarding wet-chemical processing of electrode+electrolyte bilayers, were investigated. 

In order to enable a comparison of many different processing parameters, a simple yet robust testing setup and scheme are necessary. For this study, a low but still meaningful electrode loading of 1 mAh/cm^2^ was applied to all samples. The current density was set to 0.05 mA/cm^2^ for cycling (i.e., a rate of C/20). These rather low values were selected to match the critical current density of the electrolyte layer, which was preliminarily determined by chronopotentiometry (cf. Appendix A). In addition, the low applied currents can avoid lithium plating on top of the In anodes. Indium foil was used as an anode rather than Li to avoid dendrite formation and internal short circuiting of the cells. In addition, it reduced the risk of short circuiting the cells during cell assembly since indium does not creep as much as lithium does [28]. Cycling was conducted at 10 MPa at room temperature. For comparison, only the first charge–discharge cycle is considered, as it reflects the maximum achievable specific energy and, in addition, gives information on the Li loss due to side reactions at the anode or cathode side (1st cycle coulombic efficiency). 

On this basis, different components and processing steps for composite electrodes were selected and rationally optimized.

### 3.1. Characterization of the Solid Electrolyte Powders and Film

First, the ionic conductivities of the two Li_6_PS_5_Cl powders and the prepared electrolyte film were determined by EIS measurements at different pressures. Powder pellets were prepared from the sp- and lp-Li_6_PS_5_Cl using a 16 mm diameter die. Around 350 mg were pressed at 300 MPa to yield thicknesses of approximately 1000 µm. The electrolyte film was cut with a 16 mm punch and pressed at 300 MPa. The thickness of the densified film was around 100 µm. Then, EIS measurements were conducted at increasing pressures, from 0 MPa up to 300 MPa. Similar ionic conductivities of 2∙10^−3^ S cm^−1^ were observed for both the sp-Li_6_PS_5_Cl and the lp-Li_6_PS_5_Cl materials. The prepared electrolyte film showed around half of the ionic conductivities of the pristine powders, i.e., 1∙10^−3^ S cm^−1^ (cf. Appendix A). Lp- and sp-Li_6_PS_5_Cl were further used as additives for the composite cathodes, whereas the cast electrolyte film was used as separator.

### 3.2. Active Material and Conducting Additive

The first parameters we wanted to evaluate were the shapes of the active material particles. It has been reported that the particle sizes [19] and morphologies (poly-crystalline or single-crystalline [13]) of the active cathode material have a significant impact on the electrochemical performance of solid-state batteries. It has been pointed out that the performance of e.g., single- and poly-crystalline active materials changes significantly when a solid electrolyte is used instead of a conventional liquid one. In conventional cells using organic liquid electrolytes, the contact area between the active material particles and the electrolyte changes upon cycling of the cell. Before cycling, the contact area is limited by the geometry of the active material particles. In the course of cycling, however, the particles crack, and the liquid electrolyte infiltrates the emerging free space, thus increasing the contact area and concomitantly the electrochemical performance [6,29]. For solid-electrolyte-based systems, however, cracking of the active material particles does not increase the contact area between the active material and the electrolyte but rather lowers performance [13]. 

In order to evaluate how particle morphology impacts cell performance, s-NMC811 and p-NMC811 materials were tested as conventional electrodes with an organic liquid electrolyte and as composite electrodes with the Li_6_PS_5_Cl films. First SEM measurements were conducted to elucidate the particle morphologies of the two different active materials. Large particles of spherical shape could be seen for p-NMC811, whereas smaller particles with random shapes characterize the s-NMC811 material. SEM micrographs of the two different active materials are shown in Appendix A. 

The performance in liquid electrolytes was similar to reported values [29] and the p-NMC811 performed better than the single-crystalline material (cf. Appendix A). Surface degradation of the single-crystalline materials at elevated potentials can lead to an increased charge-transfer resistance, resulting in poorer performance [29].

Next, the two active materials were tested as composite cathodes with Li_6_PS_5_Cl films. The initial composition for the composite cathodes was adopted from the literature reports [14,18,26] and was set to a relative weight ratio of AM:CA:SE = 67:3:30 (active material = AM, conducting additive = CA, solid electrolyte = SE). Initially, no binder was used, and the AM, CA, and SE were mixed manually in an agate mortar. Aligned with our expectations from the literature reports [19] a different trend was observed compared to the cells using a liquid electrolyte, see Figure 2. For s-NMC811, a lower overpotential during charging and a higher specific capacity of 161 mAh/g_NMC811_ (discharge) were achieved compared to the poly-crystalline material, for which only 93 mAh/g_NMC811_ could be obtained. The lower particle size of the s-NMC811 materials yields a higher initial contact area with the Li_6_PS_5_Cl electrolyte, and thus lower overpotentials and higher specific capacities could be obtained. This is quite evident when the impedance spectra before cycling are compared. For both s-NMC811 and p-NMC811, similar bulk resistances of around 50 Ω were observed. However, the resistance associated with interfacial reactions and contact on the cathode side shows a significant difference for the two active materials. For s-NMC811, a small additional contribution to the total resistance is observed (around 10 Ω), whereas for p-NMC811, a large additional resistance of 60 Ω can be seen. Thus, better performance can be achieved for sulfide-based solid-state batteries using s-NMC811. A coulombic efficiency of 71% for the first charge–discharge cycle was obtained, which is similar to reported values [14]. Higher coulombic efficiencies could be achieved by increasing the temperature [30] or pressure [13] during the measurements. However, the feasibility of such conditions for real-world applications is questionable, and a more detailed investigation was not possible in the scope of this work.

The next parameter we focused on was the electronic conductivity of the composite cathodes. It has been shown that for sulfide-electrolyte-containing composite electrodes the electronic conductivity can be the limiting parameter for obtaining high specific capacities [19,21]. Furthermore, conductive additives, such as C65, can catalyze side reactions of the Li_6_PS_5_Cl electrolyte with NMC cathode materials leading to lower coulombic efficiencies [18]. 

Thus, the morphology and the amount of the conductive additive need to be optimized to achieve high specific charge values. First, two different materials were tested, namely C65 and VGCF. In Figure 3, the charge/discharge behavior of s-NMC811 electrodes using sp-Li_6_PS_5_Cl and different conductive additives (either C65 or VGCF) are shown. The relative amounts of the three components were AM:SE:CA = 67:30:3 [wt./wt.] (40:55:5 [vol./vol.]). The performance of composite cathodes with C65 and VGCF is similar, although slightly lower discharge capacities were obtained with VGCF (140 mAh/g_NMC811_) than with C65 (161 mAh/g_NMC811_). Significant 1st cycle capacity losses due to increased side reactions with the sulfide electrolyte could not be observed for samples using C65, in contrast to findings reported elsewhere [18]. 

The EIS data before cycling of the cells showed similar resistance values. Compared to the C65 electrode, the cathode using VGCF showed a higher interfacial resistance. This is not quite obvious in the plots before cycling (see Appendix A), as the bulk resistance was smaller for this cell. However, after the first discharge, the bulk resistance of both cells is similar, and the higher resistance of the sample using VGCF is more evident, as shown in Figure 3.

It was also necessary to determine the required amounts of the conductive additive. Too low amounts will result in low electronic conductivity of the electrodes and thus limit rate capabilities. Too high amounts will reduce the specific energy of the cell, as the CA does not contribute to the energy storage capacity. Thus, various amounts of the two conductive additives, i.e., C65 and VGCF, have been added to composite electrodes using s-NMC811 and sp-Li_6_PS_5_Cl. They were further tested by DC polarization to determine electronic conductivity. The results are shown in Figure 4. 

Increasing the amount of conductive additive increases electronic conductivity. However, the extent of the increase is quite different in the two materials, C65 and VGCF. For VGCF, the electronic conductivity showed a considerable dependency on the applied pressure. Furthermore, even at relative amounts of up to 10 vol.%, rather low electronic conductivities of 4∙10^−4^ S cm^−1^ were obtained. In comparison, a rather low pressure dependency and values of 7∙10^−2^ S cm^−1^ were obtained for C65. Therefore, additions of VGCF to the cathode composite, especially in low relative amounts (<2 wt.%) as often encountered in the literature [18,31], do not significantly increase the electronic conductivity of the composite electrode. For C65, however, even low amounts result in a significant increase in electronic conductivity. This might be the reason for the improved electrochemical performance observed for the composite cathodes containing C65 compared to those containing VGCF (cf. Figure 2). It is assumed that C65, due to its high specific surface area and low particle size, readily forms a percolation network even at low concentrations, whereas VGCF does not due to the different morphology. Nevertheless, agglomeration of the VGCF cannot be neglected and might have contributed to the low electronic conductivities observed. In all subsequent investigations, C65 was used as conducting additive with relative amounts of 2–3 wt.% (4–5 vol%).

### 3.3. Particle Size of Sulfide Electrolyte, Active Material-to-Solid Electrolyte Ratio

The particle size of the electrolyte used in the composite cathode was reported to have a significant impact on electrochemical performance [13]. Therefore, s-NMC811 was tested with different Li_6_PS_5_Cl electrolytes. Lp-Li_6_PS_5_Cl showed large particles up to several tens of µm and above, whereas the sp-Li_6_PS_5_Cl showed smaller particle sizes of at most several µm (cf. Appendix A). Additionally, lp-Li_6_PS_5_Cl was ball-milled according to the routine found in [14], i.e., 3 g of the materials and 10 mL of p-xylene were milled at 300 rpm for 2×1 h, with a resting period of 30 min after 1 h. The reduction in particle size was confirmed by SEM and XRD measurements (see Appendix A). Composite cathodes using all three different Li_6_PS_5_Cl materials were then prepared and tested. The results are shown in Figure 5.

When using rather large electrolyte particles (i.e., lp-Li_6_PS_5_Cl), lower specific capacities could be obtained, reflecting the poor contact between active material and electrolyte particles. When smaller electrolyte particles were used (i.e., lp-Li_6_PS_5_Cl (ball-milled), sp-Li_6_PS_5_Cl), a significant improvement in the electrochemical performance of the composite cathodes could be observed. The same trend was observed in other studies [16].

Finally, the active material-to-solid electrolyte ratio has been reported to impact the performance of composite cathodes significantly [19,21]. Therefore, cathodes with a constant C65 content of 2.5 wt.% (4.5 vol.%) and different AM:SE ratios (70:30, 75:25, 80:20, 85:15, 90:10 [wt./wt.] = 40:60, 50:50, 60:40, 70:30, and 80:20 [vol./vol.]) were prepared and tested. The results are summarized in Figure 6.

The specific charge values given with respect to the active material decrease with increasing AM:SE ratios. The lower ionic conductivity of the cathode composites at high AM:SE ratios seems to decrease overall electrochemical performance. Similar findings were reported recently [16]. The lower ionic conductivity is quite evident from the increasing overpotential observed for increasing AM:SE ratios. 

The comparison of specific charge values to the entire electrode material (i.e., AM+CA+SE), though, paints a different picture. Higher amounts of SE in the cathode composite reduce the total achievable specific charge, and very similar performances can be achieved for AM:SE ratios of 40:60–60:40. Only when the content of SE is further reduced to ratios of 70:30 and 80:20, a drastic decrease in performance is observed. Thus, it seems that AM:SE ratios of 40:60–60:40 are feasible for the composite cathodes. Similar ranges are also found to work best for conventional lithium-ion batteries for which the porosities of the electrodes indicate AM:electrolyte ratios. Even for highly optimized commercial cells, porosities of around 30 vol.% [32] are employed, which reflects an AM:electrolyte ratio of 30:70. 

The reason for the lower specific charge values might also be found in the inter-particle distances of the active material. For low AM:SE ratios, i.e., high amounts of electrolytes in the composite cathodes, enough electrolyte particles are surrounding each active material particle to buffer mechanical stress from the high pressures applied during sample preparation. When less electrolyte is added to the composite, though, the NMC811 particles come in contact with each other, and densification is inhibited (similar observations are reported in [13]).

### 3.4. Influence of Binder

For cast composite cathodes, a suitable binder needs to be selected to provide good adhesion between the electrode material and the current collector. Furthermore, the selected binder should not inhibit or block the contact between the electrolyte and active material particles. SBS was selected here as binder for the cathode composites since it had already been employed successfully in other reports [14,33] and was found to inhibit ionic conductivity to a lesser degree than other binder materials, such as nitrile butadiene rubber [34].

An AM:SE ratio of 0.55 and a conductive additive (C65) and binder concentration of 2.5 wt.% (~8 vol.%) was used to investigate the influence of the binder. The results can be seen in Figure 7. Specific charge values between 212–197 mAh/g_NMC811_ and 149–135 mAh/g_NMC811_ (for AM:SE = 50:50, 60:40) were expected based on the results without binder. However, far lower values of 166 mAh/g_NMC811_ and 115 mAh/g_NMC811_ were encountered for the sample with binder. It seems that the addition of binder to the composite cathode reduces the interface between the active material and electrolyte particles and thus has an adverse effect on electrochemical performance. Consequently, lower amounts of binder (1 wt.%) should be applied to yield better performance in accordance with the literature [35]. Nevertheless, a balance between electrochemical performance and processability needs to be established, especially regarding cutting and rolling of the electrode sheets as well as further processing (e.g., mechanical stability against shear forces occurring during coating of the electrolyte slurries on top of cast electrodes).

### 3.5. Influence of Mixing Routine

Furthermore, it has been reported that the mixing routine, especially the sequence of mixing, has a pronounced effect on the achievable electrochemical performance [36]. Composite cathodes for which sequential mixing was applied, i.e., first mixing the sulfide electrolyte with the conductive additive and only then adding the active material, showed the best results. 

Accordingly, composite cathodes using s-NMC811, C65, and sp-Li_6_PS_5_Cl were prepared as described above; however, the sulfide electrolyte was first manually mixed with C65. Then, the s-NMC811 powder was added, and everything was mixed further until a homogenous mixture was obtained. The powder electrodes were further tested at 10 MPa. The results are presented in Figure 8.

A slight increase in the achievable specific charge obtained at C/20 could be seen for cathodes prepared by sequential mixing, indicating better ionic and electronic contact between active material particles.

## 4. Discussion and Conclusions

Critical parameters for the preparation of composite cathodes for sulfide-based solid-state batteries were elucidated and optimized. A summary of all composite cathode formulations is listed in Appendix A. The key findings are:

Active material—The active material for composite cathodes should have low particle sizes for a higher initial contact area with the solid electrolyte. Single-crystalline NMC811 showed both improved performance during cycling and lower impedance values compared to poly-crystalline NMC811 in accordance with the recent literature reports [13,19]. Nevertheless, concerns regarding the anisotropic volume changes in single-crystalline active material particles during de-/lithiation were raised [12].

Conductive additive: Conductive additives should be added for increased electronic conductivities of the composite electrodes. A total of 2–3 wt.% addition of C65 was enough to increase electronic conductivity by almost three orders of magnitudes. In contrast to the findings reported in [18], no significant differences in first cycle coulombic efficiencies were observed for cathodes prepared with C65 and VGCF, and thus both electronic conductive additives are eligible for the preparation of composite cathodes.

AM:SE ratio: The solid electrolyte in the cathode composite provides ion-conducting pathways and is necessary for satisfactory electrode performance. An AM:SE ratio between 40:60 and 60:40 is considered feasible and does not affect the achievable specific charge values with respect to total electrode weight. Theoretical calculations showed an ideal utilization level of the active material at an AM:SE ratio of 70:30 [21] close to the values reported here. The idealized mixing of the two components as well as the lack of any conductive additive considered in the calculations might be responsible for the difference in performance. An increase in the AM:SE ratios to higher values than 60:40 reduces the achievable specific capacities considerably, probably due to the insufficient ionic conductivity of the composite cathode [21]. For lower AM:SE ratios, higher specific charges with respect to the weight of only the active material can be reached. However, the additional weight of the SE reduces energy density when the whole electrode (i.e., AM+CA+SE) is considered. This finding highlights the importance of reporting not only the specific capacity with respect to the active material, but rather with respect to the overall composite electrode.

Binder: A polymeric binder is necessary to cast homogenous electrodes onto a current collector. Nevertheless, the addition of the binder material seems to have detrimental effects on overall cell performance in accordance with the recent literature [12,35]. Therefore, the amount of binder should be reduced to a minimum and needs to be balanced considering the resulting mechanical properties of the electrodes.

Mixing routine: It was shown that the sequence of mixing seems to have a small but significant influence on the performance of composite cathodes. Achieving a high mixing efficiency by applying different methods, such as extrusion processes, seems to be a promising way forward to increase the achievable specific capacities [37]. 

## Figures and Tables

**Figure 1 nanomaterials-13-00327-f001:**
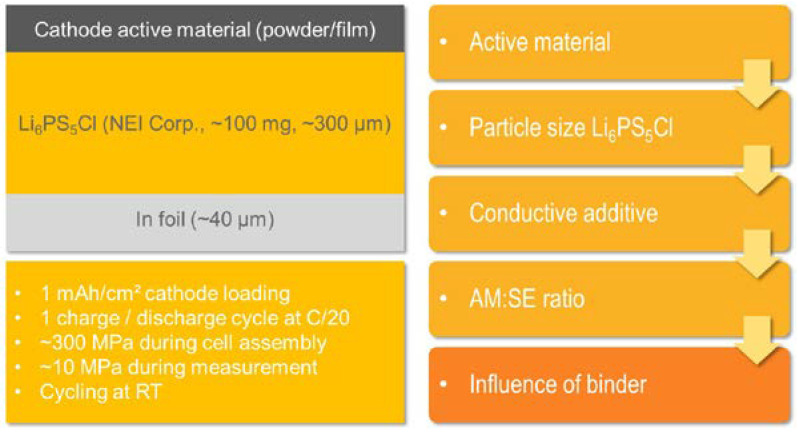
Schematic outline of the composite cathode optimization by dry- and wet-chemical processing. In addition, the testing conditions are listed.

**Figure 2 nanomaterials-13-00327-f002:**
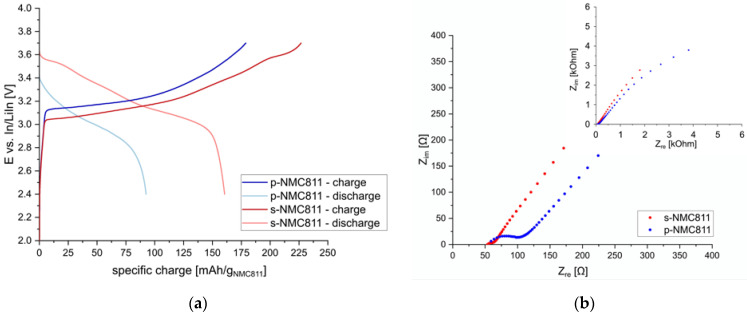
(**a**) Potential profiles of 1st charge/discharge cycle at C/20 using different NMC811 active materials; (**b**) EIS spectra before cycling (inset shows full range).

**Figure 3 nanomaterials-13-00327-f003:**
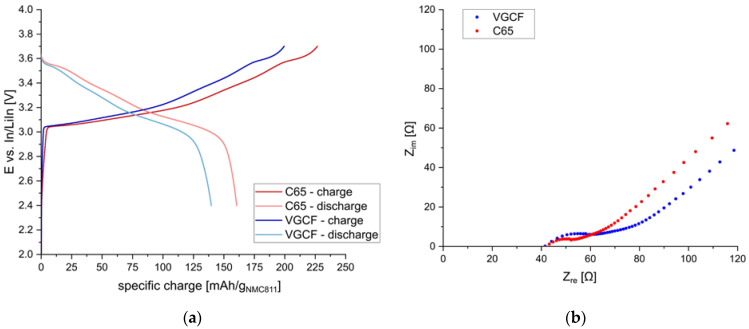
(**a**) Potential profiles of 1st charge/discharge cycle at C/20 using different conducting additives (C65, VGCF); (**b**) EIS spectra after first discharge step.

**Figure 4 nanomaterials-13-00327-f004:**
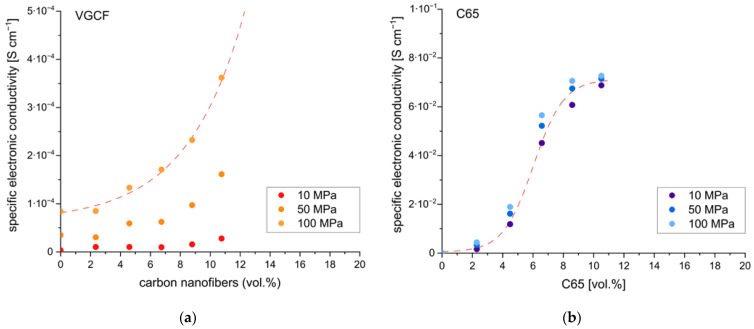
Specific electronic conductivity vs. the amount of conductive additive plots for composite cathodes using (**a**) VGCF and (**b**) C65. The dashed red lines are sigmoidal fits of the data and indicate the percolation threshold.

**Figure 5 nanomaterials-13-00327-f005:**
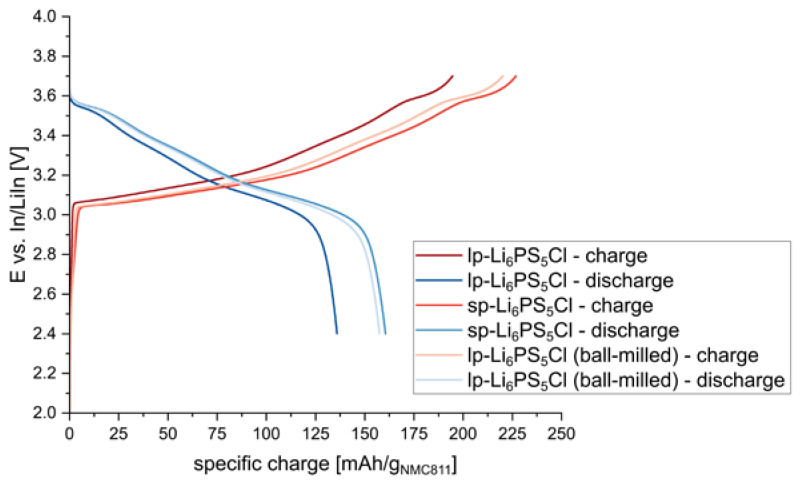
Potential profiles of 1st charge/discharge cycle at C/20 using different Li_6_PS_5_Cl materials for the composite NMC811 cathode (AM:SE:CA = 67:30:3 wt.%; 40:55:5 vol.%).

**Figure 6 nanomaterials-13-00327-f006:**
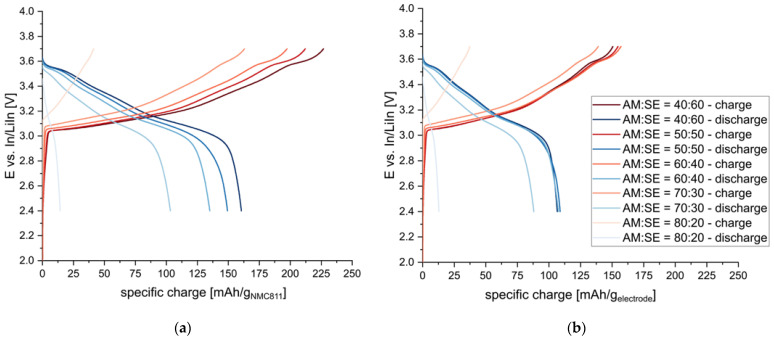
Potential profiles of composite cathodes using different AM:SE ratios (vol./vol.): (**a**) Specific charge values are given with respect to the active material and (**b**) with respect to the electrode material (AM+SE+CA).

**Figure 7 nanomaterials-13-00327-f007:**
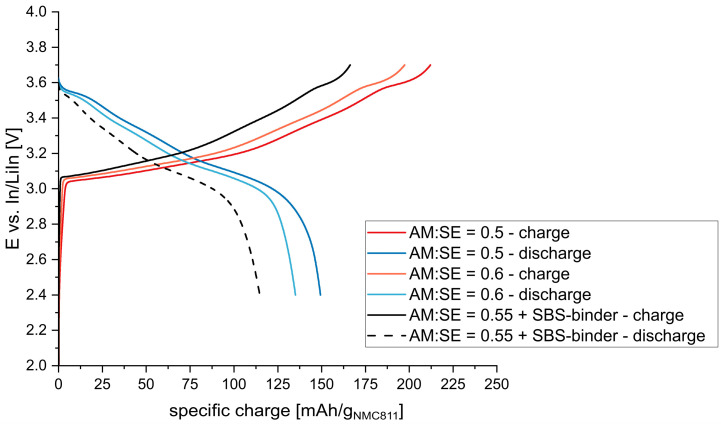
Potential profile of a cast composite cathode is compared with the profiles of two powder cathodes.

**Figure 8 nanomaterials-13-00327-f008:**
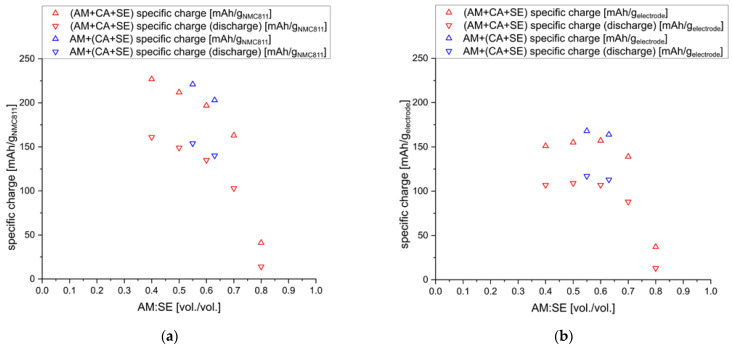
Specific charge values with respect to (**a**) the active material, (**b**) the whole electrode composite of cathodes using conventional (AM+CA+SE), and sequential mixing AM+(CA+SE) for different AM:SE ratios.

## Data Availability

Not applicable.

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
