# Peer review of "Rational Optimization of Cathode Composites for Sulfide-Based All-Solid-State Batteries"

_nanomaterials, 2023, doi:10.3390/nano13020327_

Round 1
Reviewer 1 Report
The topic of this manuscript is within the broad scope of Nanomaterials. It has current scientific interest and significant technological importance.
The work is very well presented overall, and the standard of English writing is excellent.
As far as I have been able to determine, the work is original and has sufficient novelty for publication in Nanomaterials.
The Introduction is rather too brief and under-referenced to put the present work fully into context. Indeed, the work as a whole has relatively few references, although they are recent and very relevant.
The experimental methods section provides enough detail to enable other researchers to repeat the experiments, and the results and discussion parts are satisfactory (although again I would recommend more references and literature-based discussion).
Apart from that, only minor comments are to be noted, as follows:-
1. In line 10, S cm-1 should be S cm-1.
2. For consistency, S/cm in some places should be replaced by S cm-1.
3. All abbreviations and codes (however familiar to specialists in the field) should be defined before their first use. (e.g. in line 42, C65 will not be familiar to all the readers.) In line 86, Parafilm should have an upper-case letter.
4. Although there were good practical reasons for the low current-densities used in the electrochemical experiments, the low values do limit the value of these experiments significantly. C/20 represents very mild conditions for a discharge/charge cycle, and certainly more realistic testing conditions could have been achieved over a single cycle without any risk of dendritic growth. The authors should comment on this point.
Author Response
The authors thank the reviewers for their insightful considerations and valuable comments. Below all points are addressed individually.
Reviewer #1:
regarding the general comment on the under-referenced "Introduction" and "Conclusion" parts, more references were added to underline our conclusions and set our statements into perspective.
ad 1 - In line 10, S cm-1 should be S cm-1.
has been corrected in the manuscript.
ad 2 - For consistency, S/cm in some places should be replaced by S cm-1.
has been corrected in the manuscript.
ad 3 - All abbreviations and codes (however familiar to specialists in the field) should be defined before their first use. (e.g. in line 42, C65 will not be familiar to all the readers.) In line 86, Parafilm should have an upper-case letter.
the text has been corrected and "carbon black (C65)" has been inserted instead of "C65".
ad.4 - Although there were good practical reasons for the low current-densities used in the electrochemical experiments, the low values do limit the value of these experiments significantly. C/20 represents very mild conditions for a discharge/charge cycle, and certainly more realistic testing conditions could have been achieved over a single cycle without any risk of dendritic growth. The authors should comment on this point.
In this work an electrolyte membrane has been employed rather than the pristine powder electrolyte. The critical current density of this membrane was determined experimentally as C/20 (considering an electrode loading of 1mAh/cm²). The respective potential profile has been added in the supplementary files (Figure S1) and the following text was added to the manuscript:
“These rather low values were selected to match the critical current density of the electrolyte layer, which was preliminarily determined by chronopotentiometry (cf. Figure S1). In addition, the low applied currents can avoid lithium plating on top of the In anodes.”
Reviewer 2 Report
The work presented in the current manuscript deals A detailed investigation on different key parameters for preparation of cathodes employed in all-solid-state batteries. Employing a rational approach for optimization of composite cathodes using solid sulfide electrolytes elucidated the influence of different parameters on the cycling performance. Authors have systematically carried out the experiments and presented the well-organized data. The work is interesting and can be considered for publication in Nanomaterials after addressing minor comments.
1. I recommend the authors to include some new references of Nanomaterials which are relevant to solid-state batteries to use in introduction as well as results and discussion.
2. Recheck and reformat the figures to match the standards of the Nanomaterials.
3. There are so many typographic errors throughout the manuscript as well as in figures. Authors must revise the manuscript and make it error free.
4. How is the stability of these compounds?
5. Cite the following references if authors feel they are related to their work.
https://www.mdpi.com/2079-4991/13/1/10
https://www.sciencedirect.com/science/article/abs/pii/S2352152X22023726
Author Response
The authors thank the reviewers for their insightful considerations and valuable comments. Below all points are addressed individually.
Reviewer #2:
ad 1 - I recommend the authors to include some new references of Nanomaterials which are relevant to solid-state batteries to use in introduction as well as results and discussion.
We added two relevant references from Nanomaterials, i.e. https://doi.org/10.3390/nano10081606, https://doi.org/10.3390/nano12244355
ad 2 - Recheck and reformat the figures to match the standards of the Nanomaterials
all figures have been corrected and re-uploaded
ad 3 - There are so many typographic errors throughout the manuscript as well as in figures. Authors must revise the manuscript and make it error free.
the figures, captions and the manuscript has been checked again for typographic errors
ad 4 - How is the stability of these compounds?
No work was conducted to evaluate the stability of the used compounds and thus, this matter needs to be investigated in another work. Nevertheless, it is known from literature [DOIhttps://doi.org/10.1039/D1TA09846B] that the Li6PS5Cl argyrodite is stable under inert atmosphere as well as under dry atmosphere (H2O < 100 ppm), however, it readily decomposes under ambient conditions to LiCl, Li2S, Li3PO4 and H2S.
ad 5 - Cite the following references if authors feel they are related to their work.
https://www.mdpi.com/2079-4991/13/1/10
https://www.sciencedirect.com/science/article/abs/pii/S2352152X22023726
the references which were listed do not seem to have a direct relation to the work at hand and thus were not included. Nevertheless, more relevant references were included in the “Introduction” and “Conclusion” sections.